# Quantifying uncertainties in crystal electric field Hamiltonian fits to neutron data

**Allen Scheie**

Neutron Scattering Division, Oak Ridge National Laboratory,
Oak Ridge, Tennessee 37831, USA

scheieao@ornl.gov

## Abstract

We systematically examine uncertainties from fitting rare earth single-ion crystal electric field (CEF) Hamiltonians to inelastic neutron scattering data. Using pyrochlore and delafossite structures as test cases, we find that uncertainty in CEF parameters can be large despite visually excellent fits. These results show $Yb^{3+}$ compounds have particularly large $g$-tensor uncertainty because of the few available peaks. In such cases, additional constraints are necessary for meaningful fits.

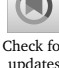

# 1 Introduction

For most magnetic systems, the single ion magnetic anisotropy [1] is crucial information: it determines not just bulk response [2], but also the strength of quantum effects [3–5], single ion magnet stability [6,7], and the exchange interactions between ions [8]. For rare earth magnetic ions, where magnetic anisotropy is strong, a common way to experimentally measure magnetic anisotropy is by fitting the crystal electric field (CEF) Hamiltonian to measured CEF excited levels. Often, this is done with neutron scattering, where the low-energy excited levels are clearly resolved [9]. However, fits to CEF neutron scattering peaks can sometimes be underdetermined (c.f. $Yb_2Ge_2O_7$ [10,11]) and CEF-derived anisotropy does not always match bulk measures of anisotropy (c.f. $YbCl_3$ [12]). Because neutron CEF studies rarely report uncertainties for the fitted CEF parameters, it is unclear how serious the discrepancies are. If CEF-derived quantities are to be useful for other studies (for example, using the $g$-tensor to fit exchange constants), the error bars for the fitted quantities must be accurately known.

In this study, we propose a method for quantifying uncertainties of CEF fits by using a stochastic search method to map out the $\chi^2$ contour. We test the method by fitting to simulated neutron scattering data for various rare earth ions. We find that the $g$-tensor uncertainties are strongly ion-dependent, with $Yb^{3+}$ often having extremely large uncertainty. These results not only demonstrate a method for rigorously defining error bars on CEF fits, but also reveal which ions are most in need of additional constraints and which quantities are most susceptible to error when fitting CEF levels.

# 2 Method

The CEF Hamiltonian can be written as:

$$\mathcal{H}_{CEF} = \sum_{n,m} B_n^m O_n^m, \tag{1}$$

where $O_n^m$ are the Stevens Operators [13,14] and $B_n^m$ are scalar CEF parameters. At a single wavevector $Q$, the neutron cross section of a crystal field Hamiltonian is written

$$\frac{d^2\sigma}{d\Omega d\omega} = A \sum_{m,n} p_n |\langle \Gamma_m | \hat{J}_\perp | \Gamma_n \rangle|^2 \delta(\hbar\omega + E_n - E_m), \tag{2}$$

where $A$ is a normalization factor, $p_n$ is the Boltzmann weight, and $|\langle \Gamma_m | \hat{J}_\perp | \Gamma_n \rangle|^2$ is computed from the inner product of the matrix element of magnetic moment with the CEF eigenstates $|\Gamma_n\rangle$. In general, because of magnetic form factors and potentially anisotropic $g$-tensors, this leads to a wavevector dependence of the intensity. However, it is common practice to fit to a constant wavevector $Q$, allowing for the $Q$-dependent terms to be neglected. In a neutron experiment, the CEF parameters $B_n^m$ are fitted to the observed intensities in Eq. 2.

To explain the method by which we determine the CEF uncertainties, we consider the example of $Yb_2Ti_2O_7$.

## Example: Pyrochlore $Yb_2Ti_2O_7$

$Yb_2Ti_2O_7$ is a pyrochlore material with magnetic $Yb^{3+}$ ions in a $D_3$ scalenohedron ligand environment, with a three-fold rotation axis along the local [111] direction shown in Fig. 1(a) [15]. In the Stevens Operator formalism [13,14], the $D_3$ symmetry gives six symmetry-allowed CEF parameters: $B_2^0, B_4^0, B_4^3, B_6^0, B_6^3, B_6^6$.

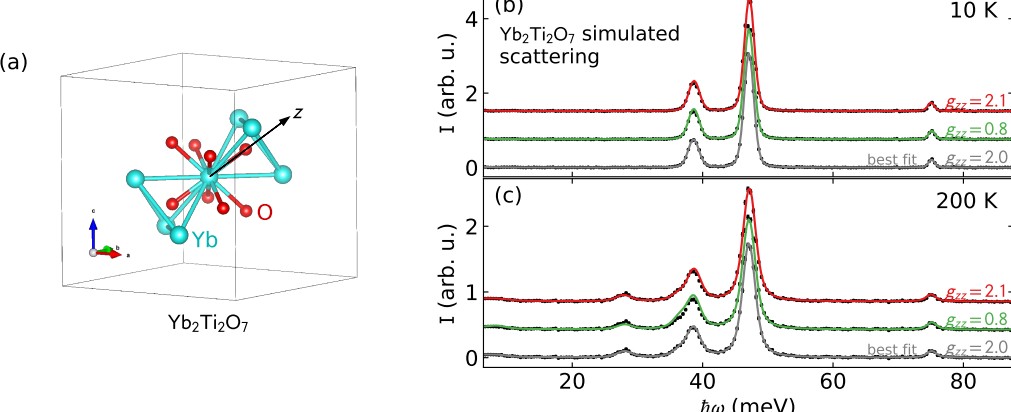

Figure 1: Pyrochlore $Yb_2Ti_2O_7$ simulated scattering and fits. (a) shows the crystal structure of $Yb_2Ti_2O_7$, with a three-fold axis along the [111] direction, which we set as $z$. (b) and (c) show the point-charge model simulated scattering at 10 K and 200 K, respectively. The three curves show three fits to the simulated scattering data, offset along the $y$ axis for clarity. The bottom (grey) shows the original model and best fit, the middle (green) shows the minimum $g_{zz}$ to within uncertainty, and the top (red) shows the maximum $g_{zz}$ to within uncertainty.

Using *PyCrystalField* software [16], we simulated CEF Hamiltonian using a point charge model based on the structure reported in Ref. [17] and 10 nearest oxygen ions. We calculated the neutron spectra at $T = 10$ K and $T = 200$ K to simulate intensities at realistic experimental temperatures. To simulate counting statistics of a real neutron experiment, we added intensity-dependent noise to the simulated data based on the Poisson counting statistics of neutron experiments, plus an intensity-independent Gaussian background noise. This method allows us to precisely define the error bar of each simulated data point. For the peak widths, we use a linear energy dependent Gaussian resolution function to define the Gaussian widths of the peaks, plus an energy independent Lorentzian broadening contribution which varies with temperature to account for finite-lifetimes at nonzero temperatures. These two broadning contributions were simulated with a Voigt profile for computational efficiency. This gave a realistic simulated neutron scattering data where the "correct" CEF Hamiltonian is exactly known.

After generating this data set, we defined a global $\chi^2$ fit function based on nine fitted parameters: the six CEF parameters, an overall scale factor, and the two Lorentzian broadening factors. The Gaussian width resolution function was fixed to the simulated values, and thus treated as precisely known. The $T = 10$ K simulation shows three peaks with three intensities, giving a total of six observable quantities related to the CEF Hamiltonian. Including a second higher temperature reveals two additional transitions: $|\Gamma_1\rangle \rightarrow |\Gamma_2\rangle$ and $|\Gamma_1\rangle \rightarrow |\Gamma_3\rangle$, bringing the total observable quantities to eight (the energies of these transitions are determined by the low-temperature peak energies, and thus contain no new information). Adding to this the two Lorentzian broadening parameters (which are not related to the CEF Hamiltonian), the total independent observed quantities in Fig. 5 comes to ten. Fitting nine parameters to ten observable quantities is a fully constrained fit. The model which best fits this simulated data, obviously, uses the nine parameters used to generate the data. This has a reduced $\chi^2_{red}$ of almost exactly 1 due to the stochastic simulated error bars. However, any solution within $\Delta\chi^2_{red} = 1$ of the best fit can be considered a valid solution to within one standard deviation uncertainty [18]. Thus, to calculate uncertainties in the CEF Hamiltonian, we must determine the allowed variation of the nine parameters such that $\Delta\chi^2 < 1$.

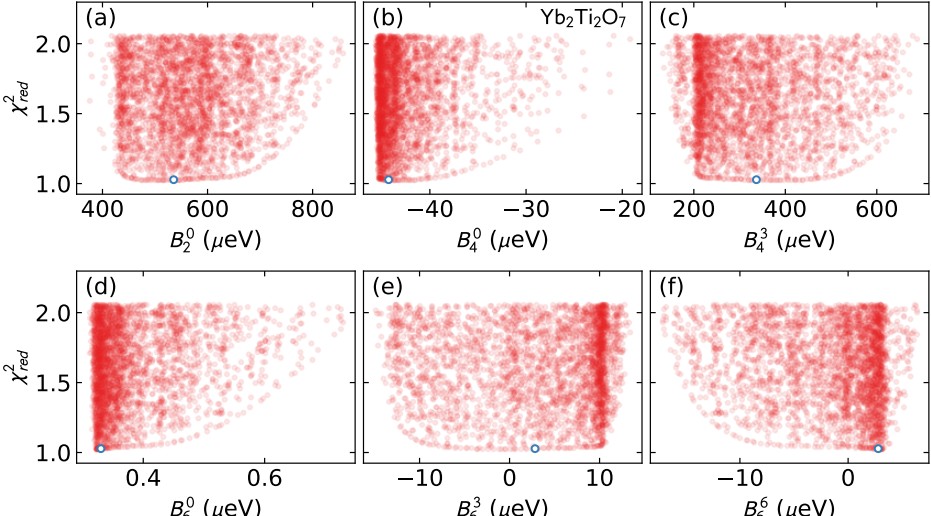

Figure 2: $\chi^2_{red}$ for the simulated $Yb_2Ti_2O_7$ data in Fig. 1. Each red point is a solution within $\Delta\chi^2 < 1$, and each panel shows a fitted CEF parameter. The "real" solution is shown with the small blue circle. The $x$ axis domain defines the uncertainty in each CEF parameter.

To calculate this, we use a two-step method: incremental search, and then Monte Carlo search. First we select a fitted parameter, fix it to a slightly increased value from the optimum fit, and fit the remaining eight parameters. If the fitted solution is less than $\Delta\chi^2_{red} = 1$, we save the solution as valid and increase the fixed parameter again, using the last fit for starting parameters. If the new solution has greater than $\Delta\chi^2_{red} = 1$ from the original optimum, we return to the optimum solution and repeat the process decreasing the fixed value from the optimum. We then repeat the process for each CEF parameter. As a second step, after looping through each CEF parameter, we then employ a series of Monte Carlo Markov Chains using each valid solution as a starting point, keeping all solutions within $\Delta\chi^2_{red} = 1$. In this way, we effectively map out the allowed variations in each parameter. The distributions of various $\chi^2_{red}$ solution for $Yb_2Ti_2O_7$ are shown in Fig. 2.

This family of $\Delta\chi^2 < 1$ solutions in Fig. 2 reveals the uncertainties in both the CEF parameters and the CEF derived quantities. The CEF parameter uncertainties are straightforward, defined by the range of parameter fit values. For derived quantities like the ground state eigenkets or the $g$ tensor, we calculate these quantities for each solution and then take the range of calculated values to be the uncertainty bounds. In this way the uncertainties are propagated through the CEF calculations.

## 3 Results

### 3.1 Pyrochlore $Yb_2Ti_2O_7$

The resulting CEF parameters with uncertainty are in Table 1, the eigenvectors and eigenvalues with uncertainty are in Table 2, and the $g$ tensor is $g_{xx} = g_{yy} = 4.10^{+0.14}_{-0.15}$, $g_{zz} = 2.05^{+0.06}_{-1.3}$.

Several things are worth noting about the $Yb_2Ti_2O_7$ CEF uncertainty calculations. First, some quantities—like $g_{zz}$—can vary quite a lot even though neutron scattering signal barely changes. To illustrate this, the maximal and minimal $g_{zz}$ models are plotted in Fig. 1(b)-(c). All these solutions would be considered "good fits" to the data (the fitted energy eigenvalues in Table 2 have tiny uncertainties), but $g_{zz} = 0.7$ is far from the true value $g_{zz} = 2.0$. Sec-

Table 1: CEF parameters for $Yb_2Ti_2O_7$ with uncertainties.

$$B_2^0 = 0.54 \pm 0.02 \qquad B_6^0 = 0.0 \pm 0.3$$
$$B_4^0 = -0.04 \pm 0.02 \qquad B_6^3 = 0.004 \pm 0.013$$
$$B_4^3 = 0.33 \pm 0.04 \qquad B_6^6 = 0.0 \pm 0.3$$

Table 2: Eigenvectors and eigenvalues for $Yb_2Ti_2O_7$ CEF Hamiltonian with uncertainties.

| E (meV) | $\lvert-\tfrac{7}{2}\rangle$ | $\lvert-\tfrac{5}{2}\rangle$ | $\lvert-\tfrac{3}{2}\rangle$ | $\lvert-\tfrac{1}{2}\rangle$ | $\lvert\tfrac{1}{2}\rangle$ | $\lvert\tfrac{3}{2}\rangle$ | $\lvert\tfrac{5}{2}\rangle$ | $\lvert\tfrac{7}{2}\rangle$ |
|---|---|---|---|---|---|---|---|---|
| 0.0 | 0.0 | -0.1(2) | 0.0 | 0.0 | -0.92(4) | 0.0 | 0.0 | 0.38(6) |
| 0.0 | 0.38(6) | 0.0 | 0.0 | 0.92(4) | 0.0 | 0.0 | -0.1(2) | 0.0 |
| 38.6(2) | 0.0 | -0.0(3) | 0.0 | 0.0 | 0.4(2) | 0.0 | 0.0 | 0.93(6) |
| 38.6(2) | 0.93(6) | 0.0 | 0.0 | -0.4(2) | 0.0 | 0.0 | -0.0(3) | 0.0 |
| 47.10(9) | 0.0 | 0.0 | 0.0 | 0.0 | 0.0 | 1.0 | 0.0 | 0.0 |
| 47.10(9) | 0.0 | 0.0 | -1.0 | 0.0 | 0.0 | 0.0 | 0.0 | 0.0 |
| 75.1(4) | -0.0(2) | 0.0 | 0.0 | -0.1(2) | 0.0 | 0.0 | -1.00(13) | 0.0 |
| 75.1(4) | 0.0 | -1.00(13) | 0.0 | 0.0 | 0.1(2) | 0.0 | 0.0 | -0.0(2) |

ond, the uncertainties in both the fitted CEF values and the resulting quantities can be highly asymmetric, evidenced both in Fig. 2 and the $g$-tensor variation.

## 3.2 Pyrochlores

The substantial variation in CEF solutions for $Yb_2Ti_2O_7$ is not so surprising given that only three peaks are visible in the low-temperature neutron spectrum. Such a fit is poorly constrained. Other ions with larger effective $J$ values would have more visible peaks, and would thus fare much better. To test this, we repeated the above method but replaced the $Yb^{3+}$ ion in $Yb_2Ti_2O_7$ point charge model with other rare earth ions: $Sm^{3+}$, $Nd^{3+}$, $Ce^{3+}$, $Dy^{3+}$, $Ho^{3+}$, $Tm^{3+}$, $Pr^{3+}$, $Er^{3+}$, and $Tb^{3+}$. The ligand environment is exactly the same for each fit—the only thing that changes is the magnetic ion. (Note that not all these materials exist as cubic pyrochlores; the point here is to compare the relative uncertainties for different ions in identical ligand environments.) The uncertainties in the ground state eigenkets and $g$ tensor values from the $\chi_{red}^2$ contours are shown in Table 3.

As expected, most other ions have smaller uncertainty in the ground state CEF wavefunction than $Yb^{3+}$. The presence of more CEF levels constrains the fit much better. (One exception to this is $Pm^{3+}$, not listed in Table 3: this ion only gives two visible peaks in its neutron spectrum, and the range of possible solutions is so great that the uncertainty was functionally infinite.) For most ions, the ground state wavefunction is generally well-constrained by a CEF fit to neutron data.

Two unusual cases here are $Sm^{3+}$ and $Ce^{3+}$, where the ground state wavefunction is exactly defined. Because of the $D_3$ symmetry of the $RE_2Ti_2O_7$ site, one eigenstate doublet is constrained to be $\lvert\pm 3/2\rangle$ exactly (this is the second $Yb^{3+}$ excited state in Table 2). For $Sm^{3+}$ and $Ce^{3+}$ in the $Yb_2Ti_2O_7$ structure, this ket is the lowest energy state. Thus, even though there is substantial variation in the $B_n^m$ values, the ground state anisotropy is precisely known. Thus a large uncertainty in the CEF parameters does not necessarily lead to a large uncertainty in the magnetic ground state.

Table 3: Uncertainties in the CEF Hamiltonian of pyrochlore $Yb_2Ti_2O_7$, but with $Yb^{3+}$ replaced with other rare earth ions. Only the three largest contributions to the ground state eigenket are listed. $Sm^{3+}$ and $Ce^{3+}$ both have a uniquely defined ground state constrained by symmetry despite variation in CEF parameters, but the rest allow for variation in the ground state wavefunction. Of all the ions, $Yb^{3+}$ and $Dy^{3+}$ have the largest $g$ tensor uncertainty. Note that many ions listed are non-Kramers and are not in general required to have a doubly-degenerate ground state, but do because of the pyrochlore lattice symmetry.

| Compound | ground state | | $g_{xx}$ | $g_{zz}$ |
|---|---|---|---|---|
| $Ce_2Ti_2O_7$ | $\psi_0+ =$ | $-1.0\lvert 3/2\rangle$ | 0.0 | 2.5714 |
| | $\psi_0- =$ | $-1.0\lvert -3/2\rangle$ | | |
| $Pr_2Ti_2O_7$ | $\psi_0+ =$ | $0.08(5)\lvert -2\rangle + 0.44(9)\lvert 1\rangle - 0.90(4)\lvert 4\rangle$ | 0.0 | $5.4^{+0.3}_{-0.2}$ |
| | $\psi_0- =$ | $0.08(5)\lvert 2\rangle - 0.44(9)\lvert -1\rangle - 0.90(4)\lvert -4\rangle$ | | |
| $Nd_2Ti_2O_7$ | $\psi_0+ =$ | $-0.29(3)\lvert -3/2\rangle + 0.04(6)\lvert 3/2\rangle - 0.956(10)\lvert 9/2\rangle$ | 0.0 | $5.80^{+0.11}_{-0.21}$ |
| | $\psi_0- =$ | $-0.29(3)\lvert 3/2\rangle - 0.04(6)\lvert -3/2\rangle - 0.956(10)\lvert -9/2\rangle$ | | |
| $Sm_2Ti_2O_7$ | $\psi_0+ =$ | $-1.0\lvert 3/2\rangle$ | 0.0 | 0.8571 |
| | $\psi_0- =$ | $-1.0\lvert -3/2\rangle$ | | |
| $Tb_2Ti_2O_7$ | $\psi_0+ =$ | $-0.031(7)\lvert -5\rangle + 0.15(5)\lvert 1\rangle + 0.988(7)\lvert 4\rangle$ | 0.0 | $11.76^{+0.12}_{-0.14}$ |
| | $\psi_0- =$ | $0.031(7)\lvert 5\rangle - 0.15(5)\lvert -1\rangle + 0.988(7)\lvert -4\rangle$ | | |
| $Dy_2Ti_2O_7$ | $\psi_0+ =$ | $0.0(3)\lvert -15/2\rangle + 0.12(5)\lvert 9/2\rangle - 0.99(12)\lvert 15/2\rangle$ | 0.0 | $19.880^{+0.099}_{-15.994}$ |
| | $\psi_0- =$ | $0.0(3)\lvert 15/2\rangle + 0.12(5)\lvert -9/2\rangle + 0.99(12)\lvert -15/2\rangle$ | | |
| $Ho_2Ti_2O_7$ | $\psi_0+ =$ | $0.040(6)\lvert 2\rangle + 0.008(4)\lvert 5\rangle + 0.9992(2)\lvert 8\rangle$ | 0.0 | $19.975^{+0.006}_{-0.008}$ |
| | $\psi_0- =$ | $-0.040(6)\lvert -2\rangle + 0.008(4)\lvert -5\rangle - 0.9992(2)\lvert -8\rangle$ | | |
| $Er_2Ti_2O_7$ | $\psi_0+ =$ | $0.105(12)\lvert -1/2\rangle - 0.24(2)\lvert 5/2\rangle - 0.962(4)\lvert 11/2\rangle$ | $0.92^{+0.06}_{-0.04}$ | $12.50^{+0.07}_{-0.04}$ |
| | $\psi_0- =$ | $-0.105(12)\lvert 1/2\rangle - 0.24(2)\lvert -5/2\rangle + 0.962(4)\lvert -11/2\rangle$ | | |
| $Tm_2Ti_2O_7$ | $\psi_0+ =$ | $0.171(5)\lvert -2\rangle + 0.17(2)\lvert 1\rangle + 0.968(4)\lvert 4\rangle$ | 0.0 | $8.65^{+0.04}_{-0.06}$ |
| | $\psi_0- =$ | $-0.171(5)\lvert 2\rangle + 0.17(2)\lvert -1\rangle - 0.968(4)\lvert -4\rangle$ | | |
| $Yb_2Ti_2O_7$ | $\psi_0+ =$ | $-0.1(2)\lvert -5/2\rangle - 0.92(4)\lvert 1/2\rangle + 0.38(6)\lvert 7/2\rangle$ | $4.10^{+0.14}_{-0.15}$ | $2.05^{+0.06}_{-1.3}$ |
| | $\psi_0- =$ | $-0.1(2)\lvert 5/2\rangle + 0.92(4)\lvert -1/2\rangle + 0.38(6)\lvert -7/2\rangle$ | | |

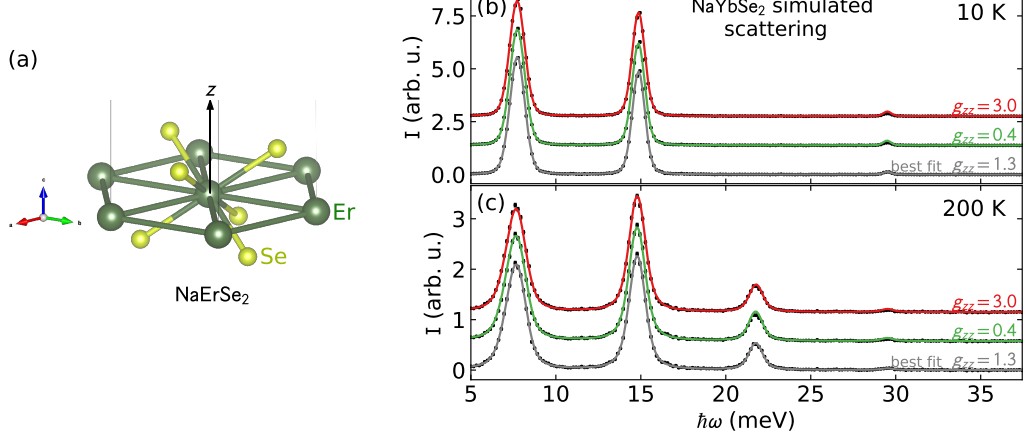

Figure 3: Delafossite $NaYbSe_2$ simulated scattering and fits. (a) shows the crystal structure of $NaErSe_2$, the basis for this fit, with a three-fold axis along the $c$ axis, which we set as $z$. (b) and (c) show the point-charge model simulated scattering with $Yb^{3+}$ as the central ion at 10 K and 200 K, respectively. The three curves show three fits: the bottom (grey) shows the original model and best fit, the middle (green) shows the minimum $g_{zz}$ to within $\Delta\chi^2 < 1$, and the top (red) shows the maximum $g_{zz}$ to within $\Delta\chi^2 < 1$.

## 3.3 Delafossites

To test whether these large error bars extend beyond pyrochlores, we now consider uncertainties in delafossite structures using the same method. The delafossite $AReB_2$ structure also has $D_3$ symmetry for the magnetic site, and for this series we based the point charge model on the $NaErSe_2$ chemical structure [19]. Thus there is the same number of fitted parameters as in the pyrochlores. The results of the fits are listed in Table 4. The simulated data and best fits for $NaYbSe_2$ are shown in Fig. 3.

Despite the different number of ligands and different environment, the results for delafossites are similar to pyrochlores: most ions have well-constrained uncertainties in the ground state anisotropy except for $Yb^{3+}$. There are however some exceptions: $Ho^{3+}$ fares poorly in the delafossite $g_{zz}$ uncertainty, while $Dy^{3+}$ fared worse in the pyrochlore $g_{zz}$ uncertainty.

The uncertainty in the $NaYbSe_2$ fitted CEF Hamiltonian becomes more interesting when we plot $\chi^2_{red}$ vs the fitted values in Fig. 4. Here there are two local minima with almost identical $\chi^2_{red}$. The "real" solution has $\chi^2_{red} = 0.9682$ and $g_{zz} = 1.316$, $g_{xx} = 3.086$ (easy plane anisotropy). The alternate solution has $\chi^2_{red} = 0.9702$ and $g_{zz} = 2.626$, $g_{xx} = 2.600$ (isotropic). This double-solution problem was encountered experimentally in $NaYbO_2$ [20]; to distinguish these two solutions would be impossible with neutron scattering data alone. In a case such as this, it is important to (i) fully map out the $\chi^2$ contour to identify competing solutions, and (ii) collect additional data or information to identify the correct anisotropy [21, 22].

As a side note, Tables 3 and 4 show candidates for strong quantum effects in pyrochlores and delafossites. The $g_{xx}$ values are directly related to $J_{\pm}$ expectation values, which are rough measures of quantum tunneling between the ground states. For pyrochlores, $Yb^{3+}$ dominates because of its large weight on $|\pm 1/2\rangle$. For the delafossites meanwhile, there are many promising candidates, most notably $Nd^{3+}$, $Ce^{3+}$, and $Yb^{3+}$ with their large $|\pm 1/2\rangle$ weights. If these

Table 4: Uncertainties in the fitted CEF Hamiltonian of delafossite materials based off the $NaErSe_2$ structure. Only ions with doublet ground states have listed $g$ tensors, some ions (Tm and Pr) have near doublet ground states with the lowest two eigenkets listed.

| Compound | ground state | | $g_{xx}$ | $g_{zz}$ |
|---|---|---|---|---|
| NaCeSe$_2$ | $\psi_0+ =$ | $-0.85(5)\|-1/2\rangle + 0.53(7)\|5/2\rangle$ | $1.85^{+0.14}_{-0.25}$ | $0.6^{+0.5}_{-0.3}$ |
| | $\psi_0- =$ | $-0.85(5)\|1/2\rangle - 0.53(7)\|-5/2\rangle$ | | |
| NaPrSe$_2$ | $\psi_0+ =$ | $0.57(2)\|-3\rangle - 0.60(4)\|0\rangle - 0.57(2)\|3\rangle$ | | |
| | $\psi_0- =$ | $0.3(2)\|4\rangle + 0.7(4)\|1\rangle - 0.6(3)\|-2\rangle$ | | |
| NaNdSe$_2$ | $\psi_0+ =$ | $0.49(4)\|-7/2\rangle - 0.55(8)\|-1/2\rangle - 0.68(4)\|5/2\rangle$ | $3.03^{+0.09}_{-0.07}$ | $0.2^{+0.2}_{-0.2}$ |
| | $\psi_0- =$ | $-0.49(4)\|7/2\rangle - 0.55(8)\|1/2\rangle + 0.68(4)\|-5/2\rangle$ | | |
| NaSmSe$_2$ | $\psi_0+ =$ | $-0.4(3)\|-1/2\rangle + 0.9(2)\|5/2\rangle$ | $0.16^{+0.44}_{-0.06}$ | $1.11^{+0.13}_{-0.89}$ |
| | $\psi_0- =$ | $-0.4(3)\|1/2\rangle - 0.9(2)\|-5/2\rangle$ | | |
| NaTbSe$_2$ | $\psi_0+ =$ | $0.33(3)\|-3\rangle + 0.88(2)\|0\rangle - 0.33(3)\|3\rangle$ | | |
| | $\psi_0- =$ | $-0.33(2)\|4\rangle + 0.88(2)\|1\rangle + 0.32(3)\|-2\rangle$ | | |
| NaDySe$_2$ | $\psi_0+ =$ | $0.51(3)\|-7/2\rangle - 0.55(2)\|5/2\rangle - 0.46(5)\|11/2\rangle$ | $8.9^{+0.3}_{-0.4}$ | $1.5^{+0.6}_{-0.6}$ |
| | $\psi_0- =$ | $-0.51(3)\|7/2\rangle + 0.55(2)\|-5/2\rangle - 0.46(5)\|-11/2\rangle$ | | |
| NaHoSe$_2$ | $\psi_0+ =$ | $-0.3(2)\|-4\rangle + 0.481(9)\|2\rangle + 0.76(6)\|5\rangle$ | $0.0$ | $7^{+2}_{-4}$ |
| | $\psi_0- =$ | $-0.3(2)\|4\rangle + 0.481(9)\|-2\rangle - 0.76(6)\|-5\rangle$ | | |
| NaErSe$_2$ | $\psi_0+ =$ | $-0.18(2)\|3/2\rangle + 0.337(15)\|9/2\rangle - 0.92(2)\|15/2\rangle$ | $0.0$ | $16.70^{+0.13}_{-2.32}$ |
| | $\psi_0- =$ | $0.18(2)\|-3/2\rangle + 0.337(15)\|-9/2\rangle + 0.92(2)\|-15/2\rangle$ | | |
| NaTmSe$_2$ | $\psi_0+ =$ | $0.663(4)\|-6\rangle - 0.219(3)\|3\rangle + 0.663(4)\|6\rangle$ | | |
| | $\psi_0- =$ | $0.680(3)\|6\rangle - 0.194(11)\|-3\rangle - 0.680(3)\|-6\rangle$ | | |
| NaYbSe$_2$ | $\psi_0+ =$ | $-0.47(5)\|-7/2\rangle + 0.5(2)\|-1/2\rangle + 0.76(9)\|5/2\rangle$ | $3.1^{+0.2}_{-0.6}$ | $1.3^{+1.7}_{-0.9}$ |
| | $\psi_0- =$ | $0.47(5)\|7/2\rangle + 0.5(2)\|1/2\rangle - 0.76(9)\|-5/2\rangle$ | | |

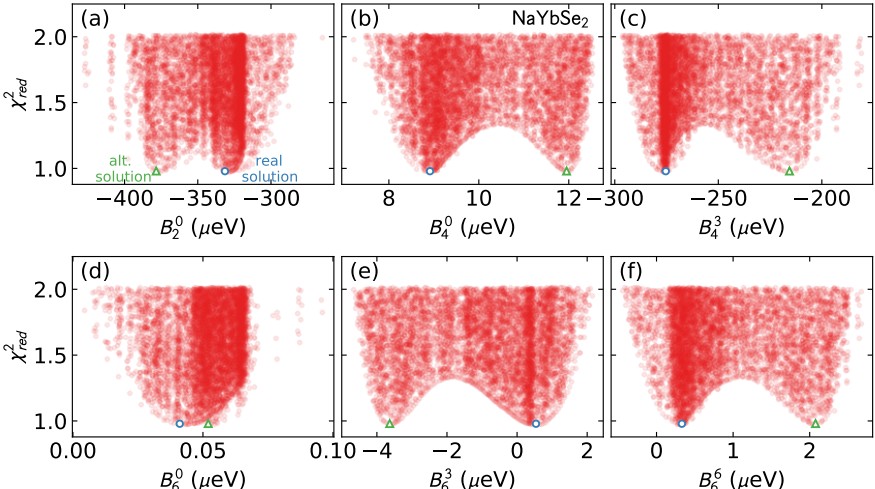

Figure 4: $\chi^2_{red}$ for the simulated NaYbSe$_2$ data in Fig. 3. Each red point is a solution within $\Delta\chi^2 < 1$, and each panel shows a fitted CEF parameter. The "real" solution is shown with the small blue circle, while the small green triangle shows an alternative solution with different anisotropy. Thus the fit is underdetermined by neutron scattering alone.

Table 5: Point charge model CEF parameters for Yb$_2$O$_3$ with uncertainties.

| | | |
|---|---|---|
| $B_2^0 = -3.3 \pm 0.2$ | $B_4^{-3} = 0.0 \pm 0.4$ | $B_4^0 = 0.0 \pm 0.8$ |
| $B_4^3 = -0.9 \pm 1.0$ | $B_6^{-6} = -0.0 \pm 0.7$ | $B_6^{-3} = -0.0 \pm 0.05$ |
| $B_6^0 = 0.0 \pm 1.11$ | $B_6^3 = -0.0 \pm 0.02$ | $B_6^6 = 0.0 \pm 0.05$ |

point charge calculations are at least approximately close to the real material Hamiltonians, these results give direction on where to find strongly quantum delafossite materials.

### 3.4 Bixbyite Yb$_2$O$_3$

In $D_3$ symmetry, the Yb$^{3+}$ ions appear to have the largest CEF uncertainties. This problem will in principle get worse as the number of crystal field parameters increases in lower symmetry structures. As an example, we considered the first Yb$^{3+}$ site in Bixbyite Yb$_2$O$_3$. This material has two symmetry inequivalent Yb$^{3+}$ sites, but we consider the first site which has $C_3$ symmetry: a three-fold axis about [111] but no mirror planes. This symmetry allows for nine CEF parameters: $B_2^0$, $B_4^{-3}$ $B_4^0$, $B_4^3$, $B_6^{-6}$ $B_6^{-3}$ $B_6^0$, $B_6^3$, and $B_6^6$.

To estimate uncertainty, we follow the same procedure outlined above. The simulated data and best fits are shown in Fig. 5, the best fit CEF parameters are in Table 5, and the calculated $g$ tensor is $g_{xx} = 1.4^{+1.5}_{-0.9}$, $g_{zz} = 7.0^{+0.4}_{-4.1}$. In this case the anisotropy is easy axis—opposite of the pyrochlore and delafossite—but as expected, the uncertainties are even larger. Indeed, the uncertanties of the CEF parameters in Table 5 are so large that most of the CEF parameters are zero to within uncertainty—hardly useful for any detailed modeling.

This is not surprising given the number of independent parameters: in this case, we are fitting 12 parameters (nine CEF parameters, a scale factor, and two Lorentzian widths) to 10 observed quantities (three transition energies, five transition intensities, and two Lorentizan widths), which technically is an underdetermined problem. That we are able to perform a fit at all is evidence that the fitted parameters are not totally independent, allowing for finite uncertainties. Thus, a Yb$^{3+}$ CEF model with nine independent parameters definitely needs more information than just neutron scattering peaks to constrain a CEF fit.

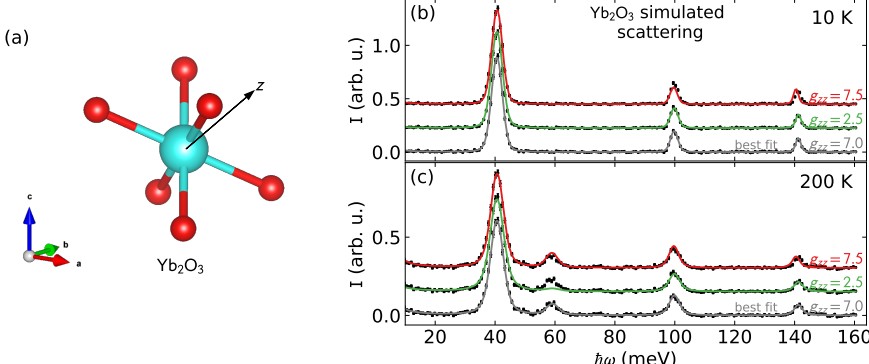

Figure 5: $Yb_2O_3$ simulated scattering and fits. (a) shows the crystal structure of $Yb_2O_3$. (b) and (c) show the point charge model simulated scattering at 10 K and 200 K, respectively. The three curves show three fits: the bottom (grey) shows the original model and best fit, the middle (green) shows the minimum $g_{zz}$ fit, and the top (red) shows the maximum $g_{zz}$ fit.

## 4 Discussion and Conclusions

The main conclusion of this numerical study is that uncertainties in CEF fits can be very large even though the fits may look visually good. This is important because these uncertanties should be propagated through other calculations that use the $g$ tensor (such as field-polarized spin wave calculations).

Furthermore, it should be noted that fits to real experimental data will probably have larger uncertainty than the idealized fits we perform here. In real experiments, there are background contributions from phonons and the sample environment, the resolution function is often not precisely known, CEF-phonon coupling affects measured intensities, and peak shapes may be asymmetric. All of these will worsen the agreement between the model and the data. Thus the true uncertainties may be much larger than those we estimate here.

This problem is particularly severe for $Yb^{3+}$ compounds as they only have three excited levels. This is unfortunate, as $Yb^{3+}$ receives much attention as an effective $J = 1/2$ host for quantum magnetism. In such cases it is necessary to include additional experimental information, like electron spin resonance as was done for $NaYbS_2$ [23], nonlinear susceptibility and high-field torsion magnetometry as was done for $CsYbSe_2$ [24], or saturation magnetization as was done for $YbMgGaO_4$ [25].

This being said, a secondary conclusion of this study is that not all calculated quantities are affected equally by CEF parameter uncertainties. The clearest examples of this are $Sm_2Ti_2O_7$ and $Ce_2Ti_2O_7$, where the ground state is precisely known despite uncertainty in the CEF model. This is also true of $Yb^{3+}$: although the pyrochlore and delafossite fits show large uncertainty for $g_{zz}$, the $g_{xx}$ is more constrained. Likewise, the $Yb^{3+}$ ground state eigenkets have relatively modest uncertainties. Therefore even if the overall anisotropy might be in question, the fitted CEF model might still give accurate and useful information about the ground state wavefunction.

In summary, we have shown by simulating and fitting to artificial CEF neutron scattering data sets that CEF fits can have very large uncertainties. In three-fold symmetric environments, $Yb^{3+}$ consistently has the largest uncertainties, highlighting the need for additional constraints when fitting its CEF levels. However, the uncertainties in calculated quantities are highly dependent upon the details of the model—some quantities are well-constrained despite uncertainty in the fitted parameters. In all cases, it is important to explore the full $\chi^2$ contour of a CEF model so that uncertainty can be known.

# Acknowledgements

The author acknowledges helpful discussions with Garret Granroth, Gabriele Sala, and Ovidiu Garlea. This research used resources at the Spallation Neutron Source, a DOE Office of Science User Facility operated by the Oak Ridge National Laboratory.

**Funding information** This manuscript has been authored by UT-Batelle, LLC, under contract DE-AC05-00OR22725 with the US Department of Energy (DOE).

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
