# Peer review of "Quantifying uncertainties in crystal electric field Hamiltonian fits to neutron data"

_SciPost Physics Core, doi:SciPost Phys. Core 5, 018 (2022)_

## Round 3 · Referee Report · Anonymous (Referee 1) · 2022-2-10

Report
Author have made a commendable effort to accommodate the reviewers' comments. However, one comment still needs further attention. This concerns the expression for the neutron scattering cross-section used for fitting, which was requested both by this reviewer and also by reviewer 2. While author has added Eq. 2, this is not yet adequate. The measured neutron intensity is determined by the matrix element of magnetic moment, not simply quantum number J. This relation involves magnetic form factor and potentially anisotropic g-tensor, which bot depend on multiplet structure and on the wave vector, Q. This needs to be properly explained. The Q-dependence is mentioned in the sentence preceding Eq. 2, which begins with, "At a single wavevector Q, ...". The Q-dependence (cf above) and how it is dealt with, need to be elaborated upon. Finally, the isotropic averaging for the matrix element of J noted in the sentence following Eq. 2 and how it is relevant for the measured cross-section should be clearly elaborated.
Requested changes
- While author has added Eq. 2, this is not yet adequate. The measured neutron intensity is determined by the matrix element of magnetic moment, not simply quantum number J. This relation involves magnetic form factor and potentially anisotropic g-tensor, which bot depend on multiplet structure and on the wave vector, Q. This needs to be properly explained. The Q-dependence is mentioned in the sentence preceding Eq. 2, which begins with, "At a single wavevector Q, ...". The Q-dependence (cf above) and how it is dealt with, need to be elaborated upon.
- The isotropic averaging for the matrix element of J noted in the sentence following Eq. 2 and how it is relevant for the measured cross-section should be clearly elaborated.

Anonymous on 2022-02-10 [id 2184]
The authors have satisfactorily responded to my comments.

---

## Round 3 · Author Response

Response to referees
Referee 1:
I thank the referee for the detailed report, and for the positive recommendation for publication. The referee requested several changes, which have been addressed as follows:
• “The manuscript lacks expressions for neutron scattering cross-section that were used to obtain fits shown in the figures. These need to be included.”
◦ Response: These are now included at the beginning of section 2.
• Because of the above, it is not clear how many independent parameters does a measurement of at most 4 peaks represent. With peak widths being fixed by the properties of the instrument, there are at most 8 independent parameters contained in 4 peaks (positions and intensities). Hence, it is obvious that a model with 9 parameters, such as author considers for Yb2O3, is over-parameterized and parameters cannot be refined. Author finds this result upon performing the fitting and chi-squared search and states it as a result of the work, "Thus, a Yb3+ CEF model with nine independent parameters definitely needs more information than just neutron scattering peaks to constrain a CEF fit." This should be revised and put in the context of relation between the number of free parameters and number of independent measured quantities contained in the crystal field peaks.
◦ Response: We have noted in the text the number of independent parameters and measured quantities for the pyrochlore fits and for the Yb2O3 fits. We have noted that the number of independent parameters for the Yb3+ measurements is the number of observed peak energies and the number of observed intensities (which is increased by the higher temperature data, when transitions from the first excited state are visible as well), and added statement to the text characterizing each fit in terms of the number of parameters.
• “More generally, the measured intensities of the 4 peaks might be related to their positions via some type of general expressions, such as the sum rules, and therefore not represent independent measurements. It is therefore not even clear whether the positions and intensities of the 4 peak do represent 6 independent measured quantities, which are needed to refine 6 crystal field parameters for pyrochlores and delafossites cases.”
◦ Response: Measurement of three peaks at low temperature and five peaks at high temperature gives eight observed quantities: three differences in energy from the ground state, and five observed intensities (the energies of the additional high temperature peaks are fully constrained by the transitions observed at lowest temperatures, but the intensities are not). The high temperature data is crucial for constraining the fit, which is why it is included.
• Explain why not use mean square deviations of the set of the obtained values for the uncertainty?
◦ Response: The uncertainty method we use is to calculate a chi^2 contour, as described in Numerical Recipes (Ref. [18]). The Monte Carlo search method is not meant to evenly sample a distribution (because we discard all points above \Delta \Chi^2 = 1), but rather to fully explore the contour where \Delta \Chi^2 = 1. In my experience, a random Monte Carlo method is the most efficient way to calculate such a contour in a high-dimensional space.
Referee 2:
I thank the referee for the detailed report, and for the positive assessment of this work’s importance. The referee raised several concerns and recommendations, which have been addressed as follows:
• “Although the equations used in the present work are known and documented in the literature, the paper would become more self-contained if the basic equations used are given, such as the Hamiltonian, the expression for the neutron scattering cross-section, and the expression for the calculation of the g-tensor. This could be done in the description of the method, in the beginning of Section 2, before the Yb2Ti2O7 pyrochlore example; the latter perhaps then requiring a minor revision.”
◦ Response: We have added the equations for the CEF Hamiltonian and the neutron cross section to the beginning of section 2.
• “In Table 2, which shows eigenvectors for different eigenvalues, it is not clear to me why two entries (lines) are given per energy (eigenvalue), as the two entries (in my understanding) are related by symmetry and the corresponding coefficients are the same and have the same uncertainties.”
◦ Response: Although the absolute value of the coefficients and their uncertainties are equal, it is not necessarily trivial to transform one degenerate Kramers eigenket to another: one must keep track of signs. It is the author’s belief that this more explicit way of presenting is more clear, especially to those less familiar with crystal field theory.
• “The same remark applies to Tables 3 and 4, where each compound has two entries, which are related by symmetry (except for the case of accidentally nearly degenerate singlets in Table 4).”
◦ Response: the same reply as above: the sign transformation is not trivial, and it is useful to state the eigenkets explicitly.
• The rational for the order of the entries in Tables 3 and 4 is unclear to me. Why not in order of the atomic number?
◦ Response: this is a good idea; the referee’s suggestion has been taken.
• In Section 4 on line 9 is said that the "peak shapes are asymmetric". This is not an intrinsic feature of crystal-field levels, but is sometimes seen on instruments on undermoderated pulsed sources with asymmetric resolution functions. I'd suggest to replace "are" by "may be"
◦ Response: the text has been changed in accord with the referee’s suggestion.
• “The importance of additional data or constraints is mentioned in the 3rd paragraph of Section 3.3. This is indeed often necessary and rather commonplace, and perhaps a few more citations in this context could guide the reader, e.g. Galera et al, J. Phys.: Condens. Matter 30 (2018) 285802 (which also outlines an alternative to the standard Monte-Carlo search), but any other relevant work would also do.”
◦ Response: A citation to Galera et al has been added. We note that we also discuss additional experimental constraints in section 4, with the examples of NaYbS2, CsYbSe2, and YbMgGaO4.
• Typos:
Page 2, 9 lines from the bottom: replace "based off" by "based on" (or "based of")
Section 4, line 11: replace "bad" by "severe" or similar.
Ref. [3]: DOI missing.
Ref. [15]: Journal name (or Editors/Publisher) missing.
Ref. [16]: Page number (356) and DOI missing.
◦ Response: we have fixed the typos, though the DOI for Ref. [16] seems to have trouble with the SciPost bibliography format which is why it does not render. I leave this as an issue for the Scipost copy editors.
Referee 3:
I thank the referee for the report and the recommendations. We have addressed them as follows:
• “the author should make clear that: i) Some of the rare-earth pyrochlores that his calculations are relevant to, the ones with light rare earths, do not exist as rare earth titanates. For example, there is no Ce2Ti2O7 or Pr2Ti2O7 or Nd2Ti2O7 as a cubic pyrochlore. The author should make this clear in the manuscript so as to avoid undue confusion. The same may be true for some of the NaErSe2 family members as well.
• ii) Some of the rare-earth pyrochlores (those based on Ho3+, Tm3+, Pr3+, and Tb3+) are non-Kramer's ions, and are not required by symmetry to have at least doubly degenerate CEF - they could have ground states that are non-magnetic singlets. There is nothing wrong with including these in the table and paper, but again it would be important to state that these results are relevant to this particular calculation.”
◦ Response: We thank the referee for the suggestions; both are valid points. We have noted in the text “(Note that not all these materials exist as cubic pyrochlores; the point here is to compare the relative uncertainties for different ions in identical ligand environments.)”.
◦ We also added the following statement to the caption for Table 3: “Note that many ions listed are non-Kramers and are not in general required to have a doubly-degenerate ground state, but do because of the pyrochlore lattice symmetry.”
Referee 1:
I thank the referee for the detailed report, and for the positive recommendation for publication. The referee requested several changes, which have been addressed as follows:
• “The manuscript lacks expressions for neutron scattering cross-section that were used to obtain fits shown in the figures. These need to be included.”
◦ Response: These are now included at the beginning of section 2.
• Because of the above, it is not clear how many independent parameters does a measurement of at most 4 peaks represent. With peak widths being fixed by the properties of the instrument, there are at most 8 independent parameters contained in 4 peaks (positions and intensities). Hence, it is obvious that a model with 9 parameters, such as author considers for Yb2O3, is over-parameterized and parameters cannot be refined. Author finds this result upon performing the fitting and chi-squared search and states it as a result of the work, "Thus, a Yb3+ CEF model with nine independent parameters definitely needs more information than just neutron scattering peaks to constrain a CEF fit." This should be revised and put in the context of relation between the number of free parameters and number of independent measured quantities contained in the crystal field peaks.
◦ Response: We have noted in the text the number of independent parameters and measured quantities for the pyrochlore fits and for the Yb2O3 fits. We have noted that the number of independent parameters for the Yb3+ measurements is the number of observed peak energies and the number of observed intensities (which is increased by the higher temperature data, when transitions from the first excited state are visible as well), and added statement to the text characterizing each fit in terms of the number of parameters.
• “More generally, the measured intensities of the 4 peaks might be related to their positions via some type of general expressions, such as the sum rules, and therefore not represent independent measurements. It is therefore not even clear whether the positions and intensities of the 4 peak do represent 6 independent measured quantities, which are needed to refine 6 crystal field parameters for pyrochlores and delafossites cases.”
◦ Response: Measurement of three peaks at low temperature and five peaks at high temperature gives eight observed quantities: three differences in energy from the ground state, and five observed intensities (the energies of the additional high temperature peaks are fully constrained by the transitions observed at lowest temperatures, but the intensities are not). The high temperature data is crucial for constraining the fit, which is why it is included.
• Explain why not use mean square deviations of the set of the obtained values for the uncertainty?
◦ Response: The uncertainty method we use is to calculate a chi^2 contour, as described in Numerical Recipes (Ref. [18]). The Monte Carlo search method is not meant to evenly sample a distribution (because we discard all points above \Delta \Chi^2 = 1), but rather to fully explore the contour where \Delta \Chi^2 = 1. In my experience, a random Monte Carlo method is the most efficient way to calculate such a contour in a high-dimensional space.
Referee 2:
I thank the referee for the detailed report, and for the positive assessment of this work’s importance. The referee raised several concerns and recommendations, which have been addressed as follows:
• “Although the equations used in the present work are known and documented in the literature, the paper would become more self-contained if the basic equations used are given, such as the Hamiltonian, the expression for the neutron scattering cross-section, and the expression for the calculation of the g-tensor. This could be done in the description of the method, in the beginning of Section 2, before the Yb2Ti2O7 pyrochlore example; the latter perhaps then requiring a minor revision.”
◦ Response: We have added the equations for the CEF Hamiltonian and the neutron cross section to the beginning of section 2.
• “In Table 2, which shows eigenvectors for different eigenvalues, it is not clear to me why two entries (lines) are given per energy (eigenvalue), as the two entries (in my understanding) are related by symmetry and the corresponding coefficients are the same and have the same uncertainties.”
◦ Response: Although the absolute value of the coefficients and their uncertainties are equal, it is not necessarily trivial to transform one degenerate Kramers eigenket to another: one must keep track of signs. It is the author’s belief that this more explicit way of presenting is more clear, especially to those less familiar with crystal field theory.
• “The same remark applies to Tables 3 and 4, where each compound has two entries, which are related by symmetry (except for the case of accidentally nearly degenerate singlets in Table 4).”
◦ Response: the same reply as above: the sign transformation is not trivial, and it is useful to state the eigenkets explicitly.
• The rational for the order of the entries in Tables 3 and 4 is unclear to me. Why not in order of the atomic number?
◦ Response: this is a good idea; the referee’s suggestion has been taken.
• In Section 4 on line 9 is said that the "peak shapes are asymmetric". This is not an intrinsic feature of crystal-field levels, but is sometimes seen on instruments on undermoderated pulsed sources with asymmetric resolution functions. I'd suggest to replace "are" by "may be"
◦ Response: the text has been changed in accord with the referee’s suggestion.
• “The importance of additional data or constraints is mentioned in the 3rd paragraph of Section 3.3. This is indeed often necessary and rather commonplace, and perhaps a few more citations in this context could guide the reader, e.g. Galera et al, J. Phys.: Condens. Matter 30 (2018) 285802 (which also outlines an alternative to the standard Monte-Carlo search), but any other relevant work would also do.”
◦ Response: A citation to Galera et al has been added. We note that we also discuss additional experimental constraints in section 4, with the examples of NaYbS2, CsYbSe2, and YbMgGaO4.
• Typos:
Page 2, 9 lines from the bottom: replace "based off" by "based on" (or "based of")
Section 4, line 11: replace "bad" by "severe" or similar.
Ref. [3]: DOI missing.
Ref. [15]: Journal name (or Editors/Publisher) missing.
Ref. [16]: Page number (356) and DOI missing.
◦ Response: we have fixed the typos, though the DOI for Ref. [16] seems to have trouble with the SciPost bibliography format which is why it does not render. I leave this as an issue for the Scipost copy editors.
Referee 3:
I thank the referee for the report and the recommendations. We have addressed them as follows:
• “the author should make clear that: i) Some of the rare-earth pyrochlores that his calculations are relevant to, the ones with light rare earths, do not exist as rare earth titanates. For example, there is no Ce2Ti2O7 or Pr2Ti2O7 or Nd2Ti2O7 as a cubic pyrochlore. The author should make this clear in the manuscript so as to avoid undue confusion. The same may be true for some of the NaErSe2 family members as well.
• ii) Some of the rare-earth pyrochlores (those based on Ho3+, Tm3+, Pr3+, and Tb3+) are non-Kramer's ions, and are not required by symmetry to have at least doubly degenerate CEF - they could have ground states that are non-magnetic singlets. There is nothing wrong with including these in the table and paper, but again it would be important to state that these results are relevant to this particular calculation.”
◦ Response: We thank the referee for the suggestions; both are valid points. We have noted in the text “(Note that not all these materials exist as cubic pyrochlores; the point here is to compare the relative uncertainties for different ions in identical ligand environments.)”.
◦ We also added the following statement to the caption for Table 3: “Note that many ions listed are non-Kramers and are not in general required to have a doubly-degenerate ground state, but do because of the pyrochlore lattice symmetry.”

---

## Round 3 · List of Changes

- Expressions for the neutron scattering cross section and the crystal field Hamiltonian are now included at the beginning of section 2.
- We have noted in the text the number of independent parameters and measured quantities for the pyrochlore fits and for the Yb2O3 fits. We have noted that the number of independent parameters for the Yb3+ measurements is the number of observed peak energies and the number of observed intensities.
- Table 3 and 4 have been reordered in order of atomic number.
- In Section 4 on line 9: "peak shapes are asymmetric" is changed to "peak shapes may be asymmetric".
- We have added a citation to Galera et al, J. Phys.: Condens. Matter 30 (2018) 285802 as another example of how to constrain crystal field fits.
- The following typos were fixed: Page 2, 9 lines from the bottom: replace "based off" by "based on" (or "based of") Section 4, line 11: replace "bad" by "severe". Ref. [3]: DOI added. Ref. [15]: Journal name added. Ref. [16]: Page number and DOI added.
- We note in the text “(Note that not all these materials exist as cubic pyrochlores; the point here is to compare the relative uncertainties for different ions in identical ligand environments.)”.
- We also added the following statement to the caption for Table 3: “Note that many ions listed are non-Kramers and are not in general required to have a doubly-degenerate ground state, but do because of the pyrochlore lattice symmetry.”

---

## Round 4 · Author Response

We thank the referee for the recommendation to clarify the Q-dependent scattering in the neutron cross section equation. Although the simulations we perform neglect such terms and thus are irrelevant to our results, it is best to avoid confusion for readers.

Sincerely,
Allen Scheie

---

## Round 4 · List of Changes

We have updated the text following Eq. 2 clarifying (i) that the cross section is based on the inner product of the magnetic moment with the eigenstates, (ii) that the equation generally has Q-dependent terms because of the magnetic form factor and potentially anisotropic g-tensors, and (iii) common experimental practice is to fit to a constant Q slice of data, such that all Q-dependent terms can be neglected.

---

## Editorial Decision

published